# Drug Sequestration in Lysosomes as One of the Mechanisms of Chemoresistance of Cancer Cells and the Possibilities of Its Inhibition

**DOI:** 10.3390/ijms21124392

**Published:** 2020-06-20

**Authors:** Jan Hraběta, Marie Belhajová, Hana Šubrtová, Miguel Angel Merlos Rodrigo, Zbyněk Heger, Tomáš Eckschlager

**Affiliations:** 1Department of Paediatric Haematology and Oncology, 2nd Faculty of Medicine, Charles University and Motol University Hospital, CZ-150 06 Prague, Czech Republic; janhrabeta@gmail.com (J.H.); mariebelhajova@gmail.com (M.B.); 2Department of Chemistry and Biochemistry, Mendel University in Brno, CZ-613 00 Brno, Czech Republic; hanasub7@gmail.com (H.Š.); merlos19792003@gmail.com (M.A.M.R.); zbynek.heger@mendelu.cz (Z.H.); 3Central European Institute of Technologies, Brno University of Technology, CZ-612 00 Brno, Czech Republic

**Keywords:** chemoresistance of cancer cells, lysosomal sequestration, V-ATPase, metallothioneins, lysosomotropic agents, V-ATPase inhibitors

## Abstract

Resistance to chemotherapeutics and targeted drugs is one of the main problems in successful cancer therapy. Various mechanisms have been identified to contribute to drug resistance. One of those mechanisms is lysosome-mediated drug resistance. Lysosomes have been shown to trap certain hydrophobic weak base chemotherapeutics, as well as some tyrosine kinase inhibitors, thereby being sequestered away from their intracellular target site. Lysosomal sequestration is in most cases followed by the release of their content from the cell by exocytosis. Lysosomal accumulation of anticancer drugs is caused mainly by ion-trapping, but active transport of certain drugs into lysosomes was also described. Lysosomal low pH, which is necessary for ion-trapping is achieved by the activity of the V-ATPase. This sequestration can be successfully inhibited by lysosomotropic agents and V-ATPase inhibitors in experimental conditions. Clinical trials have been performed only with lysosomotropic drug chloroquine and their results were less successful. The aim of this review is to give an overview of lysosomal sequestration and expression of acidifying enzymes as yet not well known mechanism of cancer cell chemoresistance and about possibilities how to overcome this form of resistance.

## 1. Introduction

Intrinsic and acquired resistance to chemotherapeutics and targeted drugs are one of the main problems in successful cancer therapy. Various mechanisms have been identified to contribute to drug resistance [1]. Resistance may be either intrinsic (primary) when the tumor cell is from the outset resistant to certain cytostatic(s) or acquired (secondary) that develops during treatment. As multiple drug resistance (MDR) we call the ability of cancer cells to resist more different anticancer drugs, which differ fundamentally in their chemical structure and mechanism of action. Many factors are known to influence the development of secondary resistance, such as the genetic changes of cancer cells, the rate of tumor growth, but also the effects of the microenvironment, e.g., hypoxia which is well known to induce chemo- and radio-resistance [2]. In the last decades, mechanisms of resistance are intensively investigated, the most well-known ones being: (i) increased drug efflux from the cancer cells or decreased drug entrance to the cancer cells; (ii) downregulation, overexpression, or modification of target molecule(s); (iii) changes in drug activating and ⁄or detoxifying enzymes; (iv) the induction of anti-apoptotic mechanisms or the inactivation of pro-apoptotic mechanisms; (v) decreased access of drug to the target cellular compartments (usually nucleus); (vi) alteration of the cell cycle (most cells move to the G_0_ phase when sensitivity to the majority of cytostatics is low); (vii) pharmacological and physiological factors, such as changes in drug metabolism and excretion, and inadequate access of the drug to the tumor including blood brain barrier in brain tumors; and (viii) also microenvironment plays a role in chemoresistance by the production of stimulating factors that induce survival pathways and it also may inactivate drugs [1,2,3,4]. In addition, a number of studies have revealed the importance of metallothioneins (MT) for the defense mechanisms of tumor cells against the effect of chemotherapy by preventing apoptosis [2].

Many experimental studies and clinical practice show that cancer cells drug resistance is caused by a combination of mechanisms and to overcome this problem will require a complex approach. There is essential for the cancer patients to receive target, individualized and combined therapy in adequate doses and intervals. This approach increases therapeutic efficacy, decreases the risk of secondary resistance and decrease tissue toxicity [3,4].

Lysosomes have been shown to trap certain hydrophobic weak base chemotherapeutics such as anthracyclines, imidazoacridinones (new antitumor compounds that inhibits topoisomerase II and intercalates to DNA), as well as some tyrosine kinase inhibitors, thereby being sequestered away from their intracellular target site [5,6]. For anticancer drugs, these targets are located either in the cell cytosol, for example heat shock proteins and microtubules, in the nucleus, i.e., DNA and topoisomerases or both in the nucleus and cytosol-like proteasome [7]. Besides, the current accepted mechanism of the role of MTs is associated with their ability to chelate a drug in combination with its enforced autophagic relocation to the lysosomal compartment, shielding the cancer cells from the cytotoxic effects of a drug [8].

The aim of this review is to give an overview of lysosomal sequestration and expression of acidifying enzymes as a mechanism of chemoresistance of tumor cells and about possibilities how to overcome this form of resistance.

## 2. Lysosomes

Lysosomes are intracellular organelles with acidic interior (pH ≤ 5), that contain acidic hydrolases and specific membrane-associated proteins, the most abundant are lysosomal integral membrane protein (LIMP)-2, lysosome-associated membrane protein (LAMP)-1, -2, and -3, and vacuolar-type H^+^-ATPase (V-ATPase). All eukaryotic cells contain lysosomes, however their morphology, size, and number are variable, depending on the cell type and their functional state [9]. Morphologically, they look like dense bodies in the cytosol with diameter usually between 0.1–2 μm and they are surrounded by single phospholipid-bilayer [10]. Lysosomal low pH, which is necessary for optimal activity of hydrolases, is achieved by the activity of the V-ATPase [11]. This protein complex composed of 13 subunits acidifies the inside of some intracellular compartments by pumping protons against their electrochemical gradient into lysosome at the expense of ATP hydrolysis [12]. Plasma membrane V-ATPases are important for invasion and migration of cancer cells as was proved by in vitro experiments with MB231 breast cancer cells transfected with V5-tagged construct of c subunit of the V-ATPase that are more invasive than control cells. On the other hand, anti-V5 antibody inhibited in vitro invasion of transfected, but not non-transfected cells. Incubation with V-ATPase inhibitor bafilomycin A also suppresses breast cancer cells invasion [13]. In vitro measurements of lysosome pH by LysoSensor and Lysosome-red fluorescent protein dyes, of cytosolic pH by fluorescent dye (2′,7′-bis-(2-carboxyethyl)-5-(and-6)-carboxyfluorescein, acetoxymethyl ester and extracellular pH using pH-sensitive microprobe in glioblastoma U87MG cells proved that bafilomycin A significantly increased lysosomal and extracellular pH and decreased cytosolic pH. This study confirmed that inhibition of V-ATPase by bafilomycin A really decreased lysosomal acidity and open new possibilities for research of lysosomal drug sequestration [14].

Lysosomal biogenesis is regulated by the coordinated lysosomal expression and regulation (CLEAR) gene network, which is activated by the main regulators E basic helix–loop–helix protein 35 (transcription factor EB, TFEB), transcription factor E3 (TFE3), and mitochondrial translational initiation factor (MTIF), upon their translocation into the nucleus. CLEAR network controls the expression of lysosomal enzymes required for the digestion of biomolecules and genes linked to the main trafficking pathways including autophagy, endo/exocytosis, and phagocytosis [9,15]. Activities of mammalian target of rapamycin (mTOR) complex-1 (mTORC1) and lysosomes are linked, since mTORC1 regulates, besides others, the expression of V-ATPase. This regulation is caused by phosphorylation of TFEB and its nuclear translocation, which is controlled by mTORC1 [16,17]. Lysosomal functions include discard of useless biomolecules and organelles, endocytosis, autophagy, exocytosis, plasma membrane repair, homeostatic maintenance of several key metabolites including cholesterol and amino acids, and apoptosis [5,9]. Lysosomal function is not only degradative, but they are important for cell signaling like in mTOR and adenylate-activated protein kinase (AMPK) pathways [18].

Moreover, lysosomes play an important role in several pathological conditions such as lysosomal storage diseases, cardiovascular diseases, Alzheimer’s disease, amyotrophic lateral sclerosis, and cancer [9,10,11].

## 3. Lysosomes, V-ATPases and Cancer

Cancer cells often show increased activities of lysosomal enzymes and changes in lysosomal volume and in their subcellular location [19,20]. There was described, that overexpression of lysosomal proteins induces the progression of cancer and metastasis and increases secretion of lysosomal proteases that are involved in tumor invasion and angiogenesis. Lysosomal proteins downregulation inhibits lysosome-mediated apoptosis. Initiator of the extrinsic apoptotic pathway, caspase-8 can induce, besides others, lysosomal deacidification and lysosomal membrane permeabilization. This lysosome-associated cell death is caused by binding of activated caspase-8 to the V0 V-ATPase domain and by blocking of its function [21]. Therefore, lysosomes are recently studied as a possible novel target for anticancer therapy. Targeting lysosomes can affect a number of cellular processes such as metabolism, reactive oxygen species (ROS) production, DNA damage, cell death, and protein secretion [18].

Accumulation of acid metabolites in cancer cells caused by Warburg’s effect (cancer cells metabolism is characterized by the suppression of oxidative phosphorylation and the activation of glycolysis as the main pathway for ATP synthesis) is cytotoxic and therefore must be detoxified [22]. This is achieved by proton pumps and transporters located in the cell and lysosomal membranes, that pump protons into the extracellular fluid and into lysosomes. They include the proton pump V-ATPase and the proton transporters: Na^+^/H^+^ exchanger (NHE1), monocarboxylate transporters (MCTs), carbonic anhydrases (CAs) particularly CA-IX, adenosinetriphosphate synthase, Na^+^/HCO_3_^−^ co-transporter and the Cl^−^/HCO_3_^−^ exchanger. Proton accumulation in lysosomes is followed by the exocytosis of their content [23]. The highly acidic cancer extracellular microenvironment induces the activation of proteases that stimulate the degradation and remolding of extracellular matrix (ECM). ECM degradation is necessary for cancer angiogenesis and metastasis. Regulation of such abnormal pH gradients by V-ATPases is important for proliferation, tumorigenesis, drug resistance, and tumor progression, and may represent a target for novel anticancer strategies [22]. The roles of lysosomes and V-ATPase in cancer cells are summarized in Figure 1.

Moreover, lysosomes play an important role in all three main types of cancer cell death i.e., apoptosis, autophagy, and necrosis. The intrinsic apoptotic pathway starts by permeabilization of the mitochondrial membrane, which leads to the release of cytochrome c. The extrinsic apoptotic pathway is initiated by external stimulation of cell death receptors. Both apoptotic pathways activate caspase cascade that is controlled by the Bcl-2 family of proteins—anti-apoptotic (e.g., Bcl-2 and Bcl-xL) and pro-apoptotic (e.g., Bax and Bid) [24]. Damaged lysosomes release proteolytic enzymes into the cytosol that initiate apoptosis [18]. Metabolism of cancer cells produce byproducts such as ammonia, ROS, and hypoxia. Autophagy plays a cytoprotective role in cancer cells by capturing damaged mitochondria that could cause apoptosis and such promotes cancer cell survival. Autophagy can have dual roles in the context of cancer as mentioned above it may have prosurvival effects and on the other hand it is also recognized as a cell death pathway. Long-lasting activation of the autophagosomal/lysosomal pathway can lead to autophagic cell death [25]. In vitro experiments show that high doses of DNA damaging cytostatics induce autophagic cell death [26]. Necroptosis is programmed necrosis, which applies in the case of apoptosis resistance caused by endogenous or exogenous factors. The receptor-interacting serine/threonine-protein kinase 1 (RIPK1), RIPK3, and Mixed Lineage Kinase Domain-Like (MLKL) have been identified as inductors of necroptosis via sphingomyelinase-mediated lysosomal membrane permeabilization (LMP). Necroptosis has also been reported to initiate autophagy by ROS production. In addition, RIPK1 modulates autophagic signaling, which is independent of necroptosis [4].

V-ATPases acidify cancer microenvironment because they are located in the plasma membrane and membranes of intracellular vacuoles—lysosomes and endosomes [23]. The sequestration and inactivation of anticancer drugs in these acidic organelles and their subsequent extrusion from the cell is one of the possible mechanisms of multidrug resistance. Overexpression of V-ATPases in several human cancers indicates that V-ATPases might be used as a therapeutic target in chemoresistant cancers [23].

Cancer progression and metastases are associated with increased pH gradient between intracellular and extracellular spaces. There was described, that reversal of this gradient is accompanied by the development of multidrug resistance [27,28]. There is supposed that this reversed gradient interferes with the permeation of weak basic drugs (such as doxorubicin and mitoxantrone) across the lipid bilayer of cells.

Those drugs are inactivated by protonation in the acid tumor microenvironment. On the other hand, weak acidic compounds, e.g., methotrexate and water-soluble drugs, such as 5-fluorouracil are resistant to acidic microenvironment [23]. An experimental study proved that V-ATPase subunit an isoform a1 is important for invasiveness of cancer cells. H-Ras-transformed cells MCF10CA1a are more invasive, as measured by in vitro Matrigel assay, than the parental MCF10a cells, this increased invasiveness was completely cancelled by V-ATPase inhibitor concanamycin A. Invasive cells have higher expression of both a1 and a3 isoforms of V-ATPase subunit a compared to the parental line. Knock-down of a3 alone or a3 and a4 together by siRNA inhibited invasiveness of MCF10CA1a cells. Overexpression of a3, but not of the other a subunit isoforms, increased the invasiveness of the parental cells [29].

Several V-ATPase inhibitors like concanamycin A, archazolid A, bafilomycin A, salicylihalamide, and NIK-12192 were shown to lead to growth arrest and cell death induction in a variety of cancer cells [30,31,32,33,34]. Nevertheless, the mode of action of V-ATPase inhibitors leading to cancer cell death is still not known and mechanisms of tumor cell invasion inhibition remain to be elucidated. It is believed, that induction of apoptosis caused by V-ATPase inhibition is due to suppression of Rab27B or activation of caspase-8 [35,36]. Rab27B is a member of Rab protein family, important regulators of vesicle trafficking, one of the mechanisms of controlling cellular functions including cell proliferation, invasion, signal transduction, and protein transport and this protein family is important for cancer development and progression [37]. V-ATPase inhibitor archazolid A induces anoikis (programmed death of cell that lost attachment to the ECM and neighboring cells) in urinary bladder and breast carcinoma cells by activating Akt and ERK in vitro and inhibits growth of mice breast cancer in vivo [38]. Administration of bafilomycin A decreases Notch signaling. In normal breast cells, V-ATPase inhibition induces accumulation of Notch in lysosomes in vitro, while in breast cancer Notch dependent cells it reduces growth. However, bafilomycin A reduces growth of T-acute lymphoblastic leukemia cell lines, although it does not affect Notch activation in those cells [39].

A number of studies conducted on patient-derived material also evidence the importance of lysosomes and V-ATPase. Huang et al. described significant positive correlation between clinical stage and grade with V-ATPase expression in samples of esophageal squamous cancer [40]. V-ATPase expression studied by immunohistochemistry in non-small cell lung carcinoma was mainly localized in the cell membrane and cytoplasm i.e., lysosomal expression. The level of V-ATPase expression was significantly lower in squamous cell lung cancer than in lung adenocarcinoma. The expression of V-ATPase positively correlated with the pathological type (higher V-ATPase expression in adenocarcinoma) and grade of non-small lung cancer and it was associated with chemotherapy drug resistance evaluated in tumor samples [41]. V1A subunit of V-ATPases was more expressed in gastric cancer samples than in normal gastric tissue, and it correlated with grade, stage, and vascular invasion. Overall survival of patients with gastric cancer whose cells express V1A subunit of V-ATPases was shorter than survival of patients suffering from cancers with negative staining [42]. Moreover, Liu et al. showed that knock down of expression of V1A subunit by siRNA decreased proliferation and invasion of gastric cancer cells in vitro [42]. Analysis of the intrinsic drug efflux capacity of leukemic cells shows that ATP Binding Cassette (ABC) transporter A3 that is localized in lysosomes is expressed in acute myeloid leukemia samples. Its higher expression was associated with shorter survival and in vitro its elevated expression induces resistance to daunorubicin, mitoxantrone, etoposide, cytosine arabinoside, and vincristine [43]. Expression of V-ATPase subunit ATP6L in hepatocellular carcinoma (HCC) was higher than in normal liver tissues and it was located both in the cytoplasm and in the plasma membrane of HCC cells. Bafilomycin A1 inhibited growth of human HCC in orthotopic xenograft model in nude mice [44]. The above-mentioned results show that higher V-ATPase expression is sign of worse prognosis. On the other hand, relationship between drug sensitivity and V-ATPase expression in renal cell carcinoma cell lines was not found, since they express many resistance factors (P-glycoprotein, gamma-glutamyl cysteine synthetase, and *cis*-diamminedichloroplatinum (II) resistance-related gene 9) [45].

Terrasi and coworkers suggested that V-ATPase subunits signature that accompanies glioma of different grade may be an independent prognostic marker of IDH wild-type low-grade glioma [46]. V1G1 subunit of V-ATPase is in many cases overexpressed in glioblastoma cells and is associated with shorter patient survival, moreover in peripheral blood of patients with glioblastoma were detected large oncosomes, a type of extracellular vesicle, that highly express V-ATPase. Authors supposed that those oncosomes could be useful as liquid biopsy [46]. Recent in silico study shows, that expression pattern of V-ATPase (combinations in up- and downregulation in the expression ratios of V-ATPase subunits and isoforms) is distinct in different cancers (leukemia, lymphoma, melanoma, myeloma, sarcoma, bladder, brain, breast, cervical, colorectal, esophageal, gastric, head and neck, kidney, liver, lung, ovarian, pancreatic, and prostate cancer) [47]. Moreover, the ratio of ATP6V1C1 and ATP6V1C2 isoforms of V-ATPase V1 subunit expression in clinical specimens may differentiate esophageal squamous cell carcinoma samples from normal surrounding tissues [47]. From the above, it is evident that the expression of individual V-ATPase subunits isoforms differs in different cancers. There is supposed that “V-ATPase profile” has significant influence on lysosomal functions. One may speculate, that this pattern of V-ATPase subunits and the expression of isoforms are expected to be implicated in the classification of some tumors.

## 4. Resistance of Cancer Cells and Lysosomes and V-ATPase

The U-A10 cells, a doxorubicin-resistant subline derived from U-937 acute monocytic leukemia cells, exhibits a redistribution of anthracyclines into enlarged lysosomes that are eccentrically placed near the nucleus. The daunorubicin distribution in cellular compartments was quantified by tritium labeled daunorubicin. It was shown that nuclei from U-A10 cells have significantly lower anthracycline accumulation than nuclei from sensitive U-937 cells. When were isolated nuclei exposed to radiolabeled drug, accumulation was similar in nuclei from both lines. On the other hand, cytoplasmic drug accumulation was higher in U-A10 cytoplasts (cell in which the nucleus has been removed) than in cytoplasts from U-937 cells. Moreover, lysosomotropic drug chloroquine decrease lysosomal anthracycline sequestration in U-A10 cells and increase daunorubicin nuclear fluorescence. That is accompanied by the restored anthracycline sensitivity of U-A10 cells. In conclusion, those experiments proved that U-A10 cells exhibit a redistribution of the lysosomal compartment and trapping of anthracyclines into expanded acidic vesicular compartment results in decreased nuclear drug accumulation and decreased anthracycline efficacy [48]. Acute promyelocytic leukemia cell line HL-60 and from it derived doxorubicin- resistant cells HL-60 ADR were used for in vitro study, which shows that weakly basic drug daunorubicin is sequestered into lysosomes, whereas sulforhodamine 101, a zwitterionic molecule (a molecule that contains an equal number of positively- and negatively charged functional group/s), is sequestered into the Golgi apparatus. Intracellular pH measurements demonstrated that in resistant cells is increased the lysosome–cytosol pH gradient. Moreover, resistant cells have higher expression of ABC transporter MRP1 localized in the Golgi apparatus. These results show two different mechanisms of intracellular sequestration in those leukemic cells [49].

Incubation of HeLa cells with hydrophobic weak base chemotherapeutics (siramesine, topotecan, sunitinib, and doxorubicin) which highly accumulate in lysosomes via cation-trapping is immediately followed by lysosomal exocytosis. Exocytosis is initiated by the transfer of lysosomes from the perinuclear zone near the plasma membrane that leads in the formation of lysosome foci in periphery and fusion of the lysosome membrane with the plasma membrane and release of the lysosomal content to the extracellular space [6]. Authors of this study assumed, that lysosome-mediated drug resistance is a two-step process: in the first step, drugs are sequestered in lysosomes and therefore are not able to achieve the target structure in the cell, and in the second step they are released from the cell by lysosomal exocytosis. Lysosomal biogenesis and exocytosis induced by lysosomal accumulation of anticancer drugs, are mediated by the release of Ca^2+^ ions from the lysosome and calcineurin (Ca^2+^-dependent serine/ threonine phosphatase) activation with consequent dephosphorylation of TFEB and its translocation to the nucleus [50]. TFEB nuclear translocation further induces lysosomogenesis [15].

Lysosomal accumulation of anticancer drugs is caused mainly by ion-trapping, but active transport of certain drugs into lysosomes by ABC transporter P-glycoprotein was also described. P-glycoprotein is located not only on the plasma membrane but also on lysosomal one, where it is also functional as was proved on cervical carcinoma derived KB-3-1 cell line and the vinblastine-resistant subline. Lysosomal P-glycoprotein increases sequestration of some of its substrates in lysosomes and this is the second mechanism of multidrug resistance caused by ABC transporter. Yamagishi and colleagues showed a decreased lysosomal sequestration of doxorubicin in *MDR1* knock-down cells [51]. Doxorubicin-resistant breast cancer cells MCF-7/ADR showed more intensive lysosomal fluorescence of doxorubicin compared to sensitive cells MCF-7.

The inhibition of ATP6L V-ATPase subunit expression by siRNA in MCF-7/ADR sensitized the cells to the cytotoxicity of doxorubicin, 5-fluorouracil, and vincristine [52]. This proved the importance of lysosomal sequestration, in which V-ATPase is significantly involved, in chemoresistance to some cytostatics. In our study we performed a comprehensive proteomic mapping and its analysis of neuroblastoma cells sensitive and resistant to cisplatin. Resistant cells overexpress ion channels transport family proteins, ATP-binding cassette superfamily proteins, solute carrier-mediated trans-membrane transporters, proteasome complex subunits, and V-ATPases. We found multiplication and enlargement of lysosomes proved by confocal microscopy and measurement of fluorescence intensity after staining by LysoTracker Red. In addition, V-ATPase inhibitor bafilomycin A sensitizes both cisplatin-resistant and sensitive neuroblastoma cells to cisplatin [53], see Figure 2.

Our study is supported by the proteomic study of Piskareva et al. that compares three pairs of neuroblastoma cell lines and from them derived cisplatin-resistant sublines. They found among other changes also different V-ATPase subunits overexpression in resistant ones [54]. Above mentioned results are consistent with the Nilsson’s study that found relationship between lysosomal pH and cisplatin sensitivity in 39 head and neck squamous cell carcinoma cell lines. Decreased expression of the V- ATPase B2 subunit was accompanied by decreasing of lysosomal acidification and sensitivity to cisplatin [55]. V0a2 subunit of V-ATPase is overexpressed on the plasma membrane and the early endosomes of ovarian cancer cells. Its inhibition sensitized resistant ovarian cancer cells to platinum drugs by acidifying of cytosol. Moreover, V0a2 expression was significantly higher in ovarian cancer tissues from drug non-responders compared to good responders [56]. In conclusion, V-ATPase play important role in resistance to cisplatin in several cancers.

We found higher protein expression of V-ATPase in ellipticine-resistant neuroblastoma cell line UKF-NB-4^ELLI^ than in the parental ellipticine-sensitive UKF-NB-4 cells. Treatment of ellipticine-sensitive UKF-NB-4 and ellipticine-resistant UKF-NB-4^ELLI^ cells induced cytoplasmic vacuolization and ellipticine was concentrated in these vacuoles see Figure 3. Confocal microscopy and staining of the cells with a lysosomal marker proved that those vacuoles are lysosomes. Transmission electron microscopy and no effect of an autophagy inhibitor wortmannin ruled out autophagy. Pretreatment with a V-ATPase inhibitor bafilomycin A or the lysosomotropic drug chloroquine prior to ellipticine enhanced the ellipticine-mediated apoptosis and decreased ellipticine-resistance in UKF-NB-4^ELLI^ cells. Moreover, pretreatment with these inhibitors increased formation of ellipticine-derived DNA adducts the most important mechanism of ellipticine anticancer effect. We concluded that resistance to ellipticine in neuroblastoma cells is associated with V-ATPase-mediated vacuolar trapping of this drug, which may be reversed by bafilomycin A and/or chloroquine [57].

Wu et al. detected lysosomal sequestration of sunitinib in dermal microvascular endothelial cells HMEC-1. Sunitinib is multiple receptor tyrosine kinases inhibitor that inhibits receptors for platelet-derived growth factor and vascular endothelial growth factor receptors, which play a role in both tumor angiogenesis and tumor cell proliferation. Sequestration was higher in endothelial cells with resistance to sunitinib induced by long-lasting incubation with low concentration of drug. Moreover, bafilomycin A and chloroquine sensitized both sensitive and sunitinib-resistant endothelial cells to sunitinib. This study shows that lysosomal sequestration may be involved in resistance to antiangiogenic therapy by targeting endothelial cells [58].

Bcl-xL- or Bcl-2-transfected small cell lung cancer cells Ms-1 are resistant to anticancer drugs (camptothecin, inostamycin, and taxol), but this resistance was reversed by adding of V-ATPase inhibitors (destruxin E, bafilomycin A, and concanamycin A). There was detected that V-ATPase inhibitors did not suppress anti-apoptotic proteins Bcl-2 nor Bcl-xL and facilitate the caspase-independent apoptotic pathway. Authors speculate that V-ATPase inhibitors may decrease anti-apoptotic protein functions by changing cellular pH and that V-ATPases are important for AIF/EndoG translocation to the nucleus that activates caspase-independent apoptotic pathway [59].

Rhabdomyosarcoma cancer stem cells (R-CSC) have lover sensitivity to doxorubicin in comparison to native cells as was detected by in vitro study. This resistance was dependent on a high expression of V-ATPase and V-ATPase inhibition by omeprazole or by siRNA enhanced doxorubicin cytotoxicity of R-CSC [60]

Ridinger et al. described that lysosomal exocytosis is dependent on histonedeacetylase 10 /HDAC10/ since selective inhibition of HDAC10 by siRNA inhibits lysosomal exocytosis, that have pro-survival effect in neuroblastoma cells treated by doxorubicin. Authors suppose that combination of HDAC10 with doxorubicin may be effective in neuroblastoma therapy [61].

V-ATPase inhibitor archazolid A induced apoptosis of leukemic cell lines and human leukemic samples. It inhibited activation of the Notch pathway, but the main mechanism of cell death induction was decreased expression of the anti-apoptotic protein surviving and depletion of iron caused by defective recycling of transferrin receptors [62].

V-ATPase subunit D1 induces Yes-associated protein (YAP) which is associated with multi-drug resistance by regulation of transcription of several genes, activation of cell proliferation, and particularly by suppression of pro-apoptotic proteins. This relationship was confirmed both in ovarian carcinoma samples obtained from patients and in vitro studies using ovarian carcinoma cell lines. Inhibition of V-ATPase restored sensitivity to paclitaxel in resistant ovarian carcinoma cells via YAP inhibition [63].

V-ATPase plays important role in epithelial–mesenchymal transition (EMT) by induction of E-cadherin internalization and recycling, and by influencing various targets as Rho-GTPase Rac1, integrin, iron metabolism, and/or Notch pathway. Knock-down of the V-ATPase and its inhibition by archazolid A blocks the EMT in in vitro model using mammary epithelial cells transduced with Twist1-ER in which was EMT induced by 4-hydroxytamoxifen [64]. EMT is accompanied by higher aggressiveness, metastases and worse response to therapy [65].

Several in vitro studies have shown that some drugs (quinacrine, chlorpromazine, amidaron, propranolol, haloperidol, chloroquine, bupivacaine, lidocaine, and imipramine) may induce an increase in lysosomal volume which may enhance lysosomal sequestration. Therefore, such drug interaction may decrease sensitivity to some drugs including anticancer ones [66,67].

## 5. Inhibition of Lysosomal Sequestration

Results of the above mentioned experiments proved that V-ATPses inhibitors or lysosomotropic drugs may restore sensitivity in resistance caused by lysosomal sequestration. From in vitro tested V-ATPase inhibitors and lysosomotropic agents (destruxin E, bafilomycin A, concanamycin A, archazolid A, and chloroquine) clinical trials in cancer have been performed only with chloroquine, a long-standing antimalarial drug, which is also used for some rheumatic diseases, and currently is being tested for the treatment of COVID-19.

Double-blind, placebo-controlled, pre-surgical window clinical study phase II in breast cancer patients shows that treatment with single-agent chloroquine 500 mg daily did not have any positive effect [68]. Phase II randomized clinical trial with gemcitabine and paclitaxel in patients with metastatic pancreatic cancer did not demonstrate longer overall survival after addition of chloroquine [69]. On the other hand, phase I study in patients with metastatic or inoperable pancreatic cancer shows that addition of chloroquine to gemcitabine was well tolerated and clinical response seemed to be promising [70]. Results of randomized, double-blind, placebo-controlled trial that examined the effect of adding chloroquine to conventional therapy for glioblastoma multiforme were not statistically significantly different but the duration of survival was more than twice longer in patients receiving chloroquine as in patients receiving placebo [71]. In phase I trial design followed by a noncomparative phase II study in glioblastoma patients after initial resection patients received chloroquine (200–800 mg oral daily) with radiotherapy and concomitant and adjuvant temozolomide any improvement of overall survival was not proved [72]. Phase I study in patients suffering from advanced solid malignancies indicates that the combination of high-dose hydroxychloroquine and temozolomide is safe and tolerable, and prolonged stable disease and partial responses in some cases proved anticancer activity of this combination in generalized melanoma [73]. From those clinical studies it can be concluded that chloroquine may improve the effect of some anticancer drugs, but a number of studies will still be needed to find the optimal combination schedule and appropriate indication.

Since lysosomal sequestration plays a significant role in resistance to anthracyclines and the pKa values of compounds correlate with level of their lysosomal sequestration Duvvuri et al. structurally modified daunorubicin to reduce its basicity. Those daunorubicin derivatives with lower pKa values showed reduced lysosomal sequestration in vitro in two resistant cell lines [74]. But these substances were not further tested as far as we know.

V-ATPase inhibition may be useful in cancer therapy also by mechanisms other than inhibition of the lysosomal sequestration. The HER2 oncoprotein targeting antibody trastuzumab is used in patients with breast and gastric cancer overexpressing HER2. However, acquired resistance to trastuzumab may develop. Endosomal recycling that is important for HER2 activity is regulated by V-ATPase. Therefore, von Schwarzenberg et al. tested V-ATPase inhibition with archazolid A in trastuzumab-resistant HER2 overexpressing breast ductal adenocarcinoma cell line JIMT-1 in vitro and in vivo. Archazolid A inhibits growth, induced apoptosis and impaired HER2 pro-survival signaling in JIMT-1 cells. JIMT-1 xenograft tumors of archazolid A treated mice had defect of HER2- recycling which decrease tumor volume compared to tumors in untreated mice. These results suggest that archazolid A inhibits growth in trastuzumab-resistant tumor cells by targeting HER2 by mechanism other than direct binding to the receptor [75].

On the other hand, lysosomes might improve efficacy of lipid nanodrugs that overcome cancer cell resistance, nanoparticles enter into the cancer cells by endocytosis and the drug is released from nanoparticles by acidic proteases in lysosomes, from which the drug can be transported into the cytoplasm by active transport, e.g., by hENT3 [76].

## 6. Metallothioneins Regulates Lysosomal Function in Cancer Chemoresistance

MTs are small cysteine-rich intracellular proteins with four major isoforms identified in mammals, designated MT-1 through MT-4. The best known biological functions of MTs are their ability to bind and sequester metal ions as well as their active role in redox homeostasis. It has been suggested that MTs provide protection against apoptosis and promote cell proliferation, leading to tumorigenesis [77].

In the last decade, it has been shown that MTs overexpression is associated with chemoresistance and poor prognosis in a variety of cancers [2]. Recently, several studies found that resistance to apoptogenic agents is common in cancers that overexpress MTs [78,79].

The role of MTs and zinc in cancer development is tightly connected. The function of MTs are strongly dependent on zinc redox state and its binding to proteins [80]. The lysosomes play a role of a “zinc sink”, working in parallel with MTs and with transport proteins. The ability to absorb zinc through active transport involving ZnT transporters makes lysosomes a good candidate for absorbing rapid cytoplasmic zinc spikes [81]. Baird et al. reported that zinc-mediated MTs up-regulation, results in lysosomal stabilization and decreased apoptosis following oxidative stress. It is suggested that the resistance against oxidative stress, known to occur in MTs-rich cells, may be a consequence of autophagic turnover of MTs, resulting in reduced iron-catalyzed intralysosomal peroxidative reactions [82]. Since these proteins are probably turned over by autophagocytosis, it may be assumed that these proteins may express their anti-apoptotic capacity by reducing intralysosomal oxidation and lysosomal membrane destabilization [82]. Besides, Ullio et al., demonstrated that MTs upregulation in combination with its enforced autophagic relocation to the lysosomal compartment is very effective in protecting rat hepatoma cells from the toxic effects of tumor necrosis factor and cycloheximide. Thus, autophagic flux actually redirects cytoplasmic MTs to the lysosomal compartment, where they will chelate intralysosomal redox-active iron, suppress lysosomal membrane permeabilization and protect cells from tumor necrosis factor and cycloheximide-induced death [8].

MT-3 is a zinc-binding protein enriched in the central nervous system and its deficiency also has a crucial role in the autophagy as well as amyloid beta endocytosis in the brain, thereby finally leading to Alzheimer’s disease as well as oxidative brain injury and cancer [83,84,85]. Our previous study indicated that lysosomal sequestration and proteasome activity may be one of the key mechanisms responsible for intrinsic chemoresistance of neuroblastoma to cisplatin [53]. Additionally, the fact that a large number of genes up-regulated in cisplatin-resistant neuroblastoma cells are involved in a vesicle-mediated transport and exocytosis, gives evidence that lysosomal enrichment and sequestration are responsible for cisplatin-chemoresistance of neuroblastoma cells [53]. In support of a role for MT-3 in lysosomal function, the absence of MT-3 results in changes in the levels of lysosomal proteins and results in reduced lysosomal degradative capacity. Lee et al., proposed that MTs might similarly stabilize lysosomes following autophagocytotic delivery to the lysosomal compartment [86]. Also, Cho et al. demonstrated the roles of MT-3 and zinc in radiation-induced autophagy and radio-resistance in glioma cells [87].

Undoubtedly, lysosomes play an important role in the heavy metal detoxification [88]. Interestingly, treatment of cancer cells with cisplatin resulted in the export of platinum via lysosomes and subsequently via exosomes [89]. The major factors involved in proteolytic degradation inside of lysosomes are proteins that belong to a class of cysteine proteases called cathepsins. Many lysosomal proteases are ubiquitously expressed, like the aspartic proteinase cathepsin D or the cysteine proteinases cathepsin B, H, and L [90]. In liver tissue, MTs proteins are degraded mainly in lysosomes by the proteases cathepsin B, and L [90]. Moreover, metal-MTs may be digested by the cysteine protease in lysosomes [91]. Therefore, these data suggest that there is evidence of the potential role for MTs in regulation of lysosomal functions and chemoresistance in cancer cells.

## 7. Conclusions and Future Directions

From the foregoing, there is no doubt that lysosomes and their enzymes, mainly V-ATPases, play an important role in the development and progression of malignant tumors. Lysosomal sequestration is one of the mechanisms of resistance to some anticancer drugs such as anthracyclines, mitoxantrone, platinum cytostatics, ellipticine, or some tyrosine kinase inhibitors like sunitinib. Moreover, V-ATPase, important lysosomal enzymes, play a significant role in resistance to other anticancer drugs, e.g., 5-fluorouracil and vinca alkaloids. Lysosomal sequestration can be successfully inhibited by lysosomotropic agents (chloroquine) and V-ATPase inhibitors (bafilomycin A, destruxin, concanamycin A, and archazolid A) in vitro. Clinical trials were performed only with one of those compounds—chloroquine and their results were less successful. The main explanation is that chemoresistance is a multifactorial event driven by multiple mechanisms, as evidenced both by experiments with chemoresistant cells prepared by long-term incubation of tumor cells with rising concentrations of antitumor drug and by comparison of samples of chemoresistant relapses with those obtained at the time of diagnosis. Another factor is the interconnection of various mechanisms, for example the localization of P-glycoprotein on both the plasma and lysosomal membranes. There was published that lysosomotropic compounds increase lysosomal pH and consequently decrease activity of lysosomal enzymes [14,49]. However, in human retinal pigment epithelial cells ARPE19 pH increase after incubation with different lysosomotropic drugs (chloroquine, fluoxetine, imipramine, dimebon, tamoxifen, chloropromazine, amitriptyline, and verapamil) was followed in the short-term by pH decrease and enlargement of lysosomes was transient [92]. This fact draws attention to the individual response of different cells to lysosomotropic substances and different dynamics of changes, so that these lysosomotropic substances can, in turn, increase lysosomal sequestration.

On the other hand, the first prerequisite for dealing with resistance is a detailed understanding of their mechanisms, which are gradually improving. In the future, for example, the use of nanoparticles and/or polymers with an anticancer drug and a chemoresistance inhibitor can be envisaged, as shown, for example, in our study combining an anthracycline cytostatic and a P-glycoprotein inhibitor. It is also a search for new anti-cancer drugs that will not induce chemoresistance as were before described daunorubicin derivatives with lower pKa [74] as well as inhibitors of various chemoresistance mechanisms. The main way, however, is to use combinations of anticancer drugs to apply different resistance mechanisms to them so that any drug remains effective. Other prerequisites are adherence to the correct dose and administration interval of anticancer drugs.

Thus, it can be concluded that chemoresistance is a complex process and lysosomal sequestration plays an important role in it. Clinically useful inhibitors are not yet available, although they are subject to intensive research. Even the best studied P-glycoprotein inhibitors have not been applied in practice.

## Figures and Tables

**Figure 1 ijms-21-04392-f001:**
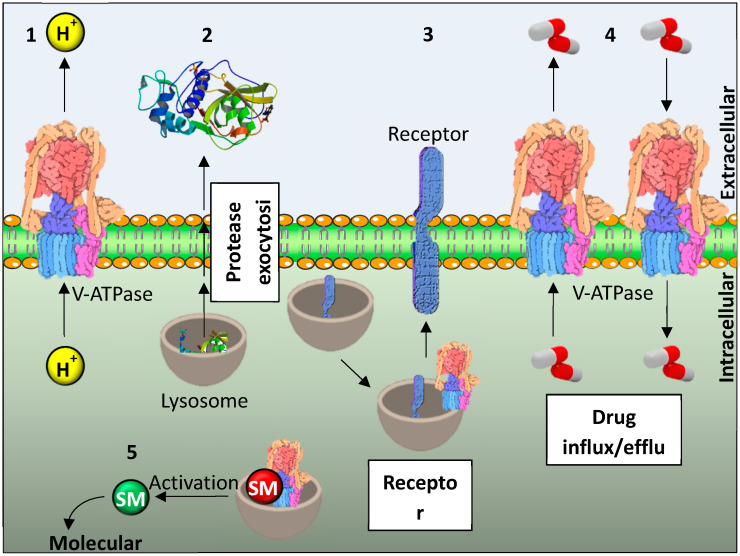
The roles of lysosomes and V-ATPase in cancer cells. (**1**) Protons produced by glycolysis (Warburg’s effect) are transported across the membrane from the cell by V-ATPase which prevents intracellular acidosis. (**2**) Proteases are activated in lysosomes and secretory vesicles and transported from the cell by exocytosis. Low extracellular pH in cancer microenvironment is optimal for their activity. Proteases are important for cancer cells invasiveness and angiogenesis. (**3**) Lysosomes and V-ATPase are important for receptor recycling. (**4**) Lysosomes are important for influx and efflux of different drugs. (**5**) Lysosomes and V-ATPase process signal molecules that regulate signal pathway. SM—signal molecule. The structures used in figure are vacuolar ATPase [Protein Data Bank (PDB) entry 5vox] and human cathepsin K (PDB entry 5J94).

**Figure 2 ijms-21-04392-f002:**
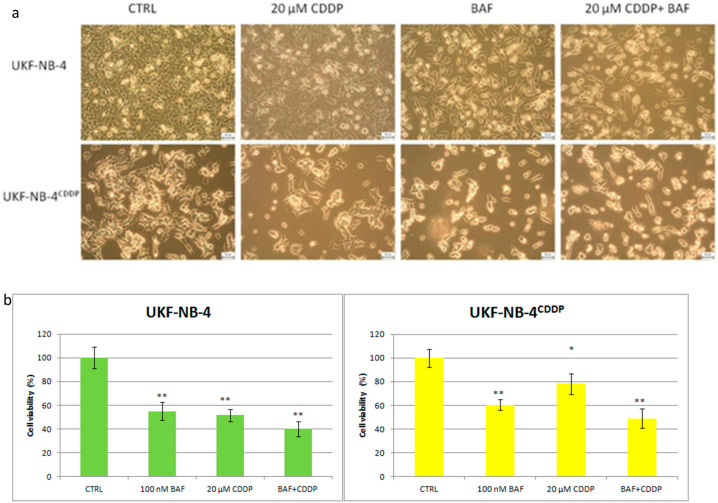
Detection of cellular viability by morphology-original magnification 100× (**a**) and Alamar Blue- (**b**) both in UKF-NB-4 sensitive and cisplatin-resistant UKF-NB-4^CDDP^ neuroblastoma cells treated with 100 nM bafilomycin A, 20 μM cisplatin, or a combination of both for 24 h. * *p* < 0.05 ** *p* < 0.01 CTRL- control, BAF- bafilomycin A, CDDP- cisplatin.

**Figure 3 ijms-21-04392-f003:**
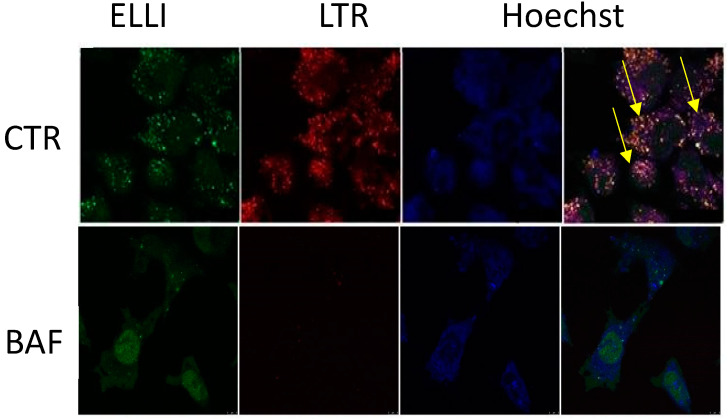
Confocal microscope images demonstrate co-localization (yellow) of ellipticine (green) and LysoTracker (red), (marker of the acidic lysosomal compartment) in UKF-NB-4 cells. Ellipticine is present (sequestrated) in lysosomes. Cells were incubated with ellipticine with or without bafilomycin A (BafA) Pretreatment of the UKF-NB-4 cells with bafilomycin A prior to ellipticine decreased amounts of formed vacuoles. Nuclei were stained with Hoechst 33342 (Hoechst). Yellow arrows- co-localization of ellipticine a LysoTracker. Original magnification 1000×. Photo M. Belhajova, J. Hrabeta.

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
