# Peer review of "Drug Sequestration in Lysosomes as One of the Mechanisms of Chemoresistance of Cancer Cells and the Possibilities of Its Inhibition"

_ijms, 2020, doi:10.3390/ijms21124392_

Round 1
Reviewer 1 Report
In this review the authors describe the role of lysosomes in cancer, highlighting their important role in chemoresistance
- The authors should add a figure describing the mechanisms explained in the section 3.
- I am wondering if any difference in lysosome behavior was found based on the type of tumor according to the authors?
- The same for drugs. Are there drugs more exposed to the lysosomal sequestration?
- What could be the role of lysosomes in cancer stem cells?if there is a possible role according to the authors.
Author Response
Thanks to the reviewer for comments.
The authors should add a figure describing the mechanisms explained in the section 3. - We add figure that summarized the roles of lysosomes and V-ATPases in cancer cells.
I am wondering if any difference in lysosome behavior was found based on the type of tumor according to the authors? We explain in more detail differences of V-ATPases isoforms and subunits expression in different tumor. There is supposed that “V-ATPase profile” has significant influence on lysosomal functions.
The same for drugs. Are there drugs more exposed to the lysosomal sequestration? We mentioned in conclusions resistance to which drugs may be caused by lysosomal sequestration.
What could be the role of lysosomes in cancer stem cells?if there is a possible role according to the authors. We add one paragraph about V-ATPase and chemoresistance of rhabdomyosarcoma CSC.
Reviewer 2 Report
This is an interesting to read review on a somewhat overlooked topic of cancer chemotherapy resistance. I learned a lot from reading it.
Line 56: Re-word sentence this sentence. In its present form it is not a very elegant read.
Author Response
Thanks to the reviewer for comments.
Line 56: Re-word sentence this sentence. In its present form it is not a very elegant read. We fully agree with the objection and we reformulated this sentence.
Round 2
Reviewer 1 Report
The manuscript has been significantly improved